# Molecular Basis of Cardiomyopathies in Type 2 Diabetes

**DOI:** 10.3390/ijms25158280

**Published:** 2024-07-29

**Authors:** Silvia Giardinelli, Giovanni Meliota, Donatella Mentino, Gabriele D’Amato, Maria Felicia Faienza

**Affiliations:** 1Department of Medical Sciences, Pediatrics, University of Ferrara, 44121 Ferrara, Italy; silvia.giardinelli@edu.unife.it; 2Department of Pediatric Cardiology, Giovanni XXIII Pediatric Hospital, 70126 Bari, Italy; giovanni.meliota@gmail.com; 3Pediatric Unit, Department of Precision and Regenerative Medicine and Ionian Area, University of Bari “Aldo Moro”, 70124 Bari, Italy; donatella.mentino@uniba.it; 4Neonatal Intensive Care Unit, Di Venere Hospital, 70012 Bari, Italy; gabriele.damato@asl.bari.it

**Keywords:** type 2 diabetes, diabetic cardiomyopathy, heart failure, cardiac remodeling, epigenetics, molecular mechanisms, Glucagon-like Peptide-1 (GLP-1) receptor agonists, gliflozins, therapeutic approaches

## Abstract

Diabetic cardiomyopathy (DbCM) is a common complication in individuals with type 2 diabetes mellitus (T2DM), and its exact pathogenesis is still debated. It was hypothesized that chronic hyperglycemia and insulin resistance activate critical cellular pathways that are responsible for numerous functional and anatomical perturbations in the heart. Interstitial inflammation, oxidative stress, myocardial apoptosis, mitochondria dysfunction, defective cardiac metabolism, cardiac remodeling, hypertrophy and fibrosis with consequent impaired contractility are the most common mechanisms implicated. Epigenetic changes also have an emerging role in the regulation of these crucial pathways. The aim of this review was to highlight the increasing knowledge on the molecular mechanisms of DbCM and the new therapies targeting specific pathways.

## 1. Introduction

Type 2 diabetes mellitus (T2DM) is a metabolic disorder characterized by hyperglycemia resulting from peripheral insulin resistance, failure of pancreatic beta cells or both [1]. The prevalence of T2DM is rising among adolescent population in many countries, although type 1 diabetes mellitus (T1DM) still has the highest prevalence [2]. This increase in T2DM in youths has been attributed to the rising prevalence of obesity worldwide. Excess adipose tissue and especially the accumulation of visceral fat predispose individuals to insulin resistance and T2DM [3]. Moreover, T2DM is strictly associated with increased morbidity and mortality, mainly due to long-term vascular complications.

Cardiovascular (CV) risk factors appear early in the clinical course of T2DM [4]. Children and adolescents may already show subclinical ventricular abnormalities, and insulin resistance can negatively impact on endothelial function since childhood, particularly in children born small for gestational age [5,6]. Numerous evidence suggests that early-onset T2DM causes an increased risk of CV mortality [7,8,9,10]. Furthermore, the coexistence of obesity worsens the CV risk of T2DM individuals [11].

Diabetic cardiomyopathy (DbCM) is defined by “the existence of abnormal myocardial structure and performance in the absence of other cardiac risk factors such as coronary artery disease, hypertension, and significant valvular disease, in individuals with diabetes mellitus” [12]. The prevalence of DbCM is approximately 16% in adult individuals with T2DM [13]. Heart failure is the most common T2DM complication, with a prevalence of 3.4% in adults [14].

The course of DbCM has two phases: an early stage, in which the main features are left ventricular (LV) hypertrophy and impaired diastolic function, and a later one, in which myocardial fibrosis and systolic dysfunction are observed. The exact mechanisms of DbCM are still not fully understood, especially in youth, although chronic hyperglycemia and insulin resistance with compensatory hyperinsulinemia exert toxic effects on the myocardium through direct and indirect pathways [15]. Several factors, such as inflammation, deposition of advanced glycation end products (AGEs), oxidative stress, mitochondrial dysfunction, lipotoxicity, and alterations of subcellular components, contribute to numerous functional and anatomical perturbations [16,17,18]. The aim of this narrative review was to highlight the most recent evidence on the molecular basis of pathogenetic mechanisms of cardiomyopathies in T2DM patients and the future therapeutic prospectives.

## 2. Inflammation

Chronic low-grade inflammation is a key component in the course of T2DM and its related complications, which contributes to peripheral disease progression and leads to structural and functional changes in cardiomyocytes [17]. This condition is mediated by cytokines that are primarily released via the activation of nuclear factor NF-κB through different mechanisms: the most relevant are represented by hyperglycemia, increased levels of fatty acids (FAs), mitochondrial dysfunction, increased reactive oxygen species (ROS), derangement in the renin–angiotensin–aldosterone system (RAAS) and the accumulation of AGEs [19]. AGEs are proteins or lipids that become glycated after exposure to glucose and accumulate in many different cell types. They act via the AGE receptor (RAGE) triggering the activation of NF-κB signaling, as shown in murine models in which diabetes increased cytosolic levels of ROS, RAGEs, Tumor Necrosis Factor alpha (TNF-α) and nuclear levels of NFkB-p65 in cardiomyocytes [20]. Indeed, NF-κB activation promotes an upregulation of pro-inflammatory cytokines, such as TNF-α, Interleukin 1 β (IL-1β) and IL-6 (Figure 1) [21].

Proinflammatory cytokines causes damage to mitochondria and the endoplasmic reticulum (ER) and generate ROS and reactive nitrogen species (RNS), and thus, contribute to apoptosis, autophagy and pyroptosis, which is an inflammation-induced cell death [22,23]. The increased ROS production causes the activation of the NLR family pyrin domain containing 3 (NLRP3) inflammasome, which through caspase-1, promotes the release of IL-1β-mediated interactions [19]. Luo B. et al. explored the central role of NLRP23 in DbCM. In their study, the activation of NLRP3 inflammasome was associated with worse cardiac function in mice models of streptozotocin (STZ)-induced diabetes due to IL-1β release and caspase-1 activation. In high-glucose-treated H9c2 cardiomyocytes, increased ROS activation and phosphorylation of NF-kB p65 were found, suggesting their role in activating NLRP3 inflammasome. Furthermore, the downregulation of NLRP3 suppressed the release of IL-1β, IL-18, TNF-α and IL-6; reduced the rate of apoptosis; and inhibited caspase-1 activation, with the consequence of reduced pyroptosis. Furthermore, the inhibition of NF-kB caused reduced ROS production and NLRP3 expression, suggesting that NF-kB can be a link between ROS and NLRP3 [24].

An increased expression of TNF-α can induce the aggregation and infiltration of immune cells (e.g., macrophages, T cells, B cells, neutrophils, dendritic cells and mast cells), and damage associated molecular pattern molecules (DAMPs), which are contributing factors in determining myocardial injury in other cardiac diseases [25]. Increased IL-1β production inhibits the beta-adrenergic receptor, consequently causing decreased myocardial contractility [26]. Overall, cytokines released via the NF-κB pathway contribute to cardiomyocytes stress and damage, causing consequent remodeling.

Lower levels of adiponectin have been reported in individuals with diabetes and heart failure. Adiponectin has protective properties through the modulation of endothelial adhesion molecules, such as vascular cell adhesion protein-1 (VCAM-1), intracellular adhesion molecule-1 (ICAM-1) and E-selectin, and through the inhibition of TNF-α and NF-κB [27]. Adiponectin also enhances glucose uptake into skeletal muscle and restrains hepatic gluconeogenesis [28]. Adiponectin deficiency causes mitochondrial dysfunction in various organs, including the heart, and in obese children, a worse cardiovascular risk profile has been related to lower levels of adiponectin [29,30]. Recently, high glucose levels have been linked to an increase in the expression of miR-223-3p in cardiomyocytes, causing an increased release of pro-inflammatory cytokines and subsequent pyroptosis [31].

Transcriptomics studies demonstrated an upregulation in genes involved in the inflammatory pathways in cardiac tissues isolated from T2DM mice, and particularly inflammasome-related genes (e.g., Aim2, Nlrc4) associated with antigen presentation and langerin, which is an antigen uptake receptor most often expressed by Langerhans cells and stimulates T cell responses [32,33,34]. Inflammation and metabolic stress contribute to the senescence of both cardiomyocytes and immune cells, which causes impaired energetic and metabolic perturbations and ultimately contributing to cardiac dysfunction [35]. Overall, ROS production is increased and protection mechanisms are decreased in a diabetic heart: ROS induce oxidative damage to DNA, proteins and lipids, and may inhibit autophagy, which is a regulated process that removes dysfunctional cellular components, and mitochondria dysfunction causes energy imbalance in the heart [36].

## 3. Mitochondrial Dysfunction

In individuals with T2DM, defects in the mitochondrial structure and function contribute to the development of insulin resistance in skeletal muscle, adipose tissue and pancreatic β-cells [37,38]. Mitochondria dysfunction appears in the early stages of DbCM development, which induces damage through increased ROS generation and reduced ATP production, which can lead to cardiovascular diseases [39,40,41]. Damaged mitochondria cause mtDNA release with activation of the cGAS-STING pathway, which, in turn, triggers myocardial cell apoptosis and an inflammatory response, contributing to the development of myocardial cell hypertrophy and cardiac remodeling [42].

The heart has a high energy request, consuming approximately 8% of the total ATP expenditure of the body. In the heart, a higher mitochondrial volume density is necessary to produce ATP via oxidative phosphorylation, mainly from the oxidation of FAs in physiologic conditions [43,44]. In T2DM, the rise in serum FAs and triglyceride levels promotes higher rates of FAs uptake and oxidation, with lower glucose utilization [45,46]. This is driven by cardiac-specific overexpression and the increased activity of peroxisome proliferator-activated receptor α (PPARα) [47,48]. Increased rates of fatty acid oxidation (FAO) and the shift to non-oxidative metabolism have different effects: First, if the delivery of FA to the mitochondria exceeds their capacity, acyl-CoAs accumulate and they are shifted to alternative pathways of lipid metabolism, with the production of other products. Among them, ceramides and diacylglycerols accumulate and exert cardiac toxicity, which activates protein kinase C signaling, apoptosis, ER stress and increased ROS production (Figure 1) [49,50,51,52].

Increased rates of FAO also decrease the mitochondrial efficiency of ATP regeneration and increase myocardial O_2_ consumption, with a total effect of decreased cardiac efficiency.

Elevated FA levels in diabetes can also impair cardiac Hypoxia-inducible factor 1-alpha (HIF-1α) accumulation, with a subsequent reduced cardiac adaptation to hypoxic conditions based on a model of insulin resistance in HL-1 cardiomyocytes, which affects cardioprotection [53].

It was also proposed that increased levels of mitochondrial calpains or monoamine oxidase (MAO) can have a role in inducing mitochondria dysfunction [54]. Furthermore, in a diabetic heart, the reduction in levels of the mitochondrial deacetylase SIRT3 promotes acetylation and, consequently, inactivation of Manganese superoxide dismutase (MnSOD), which is a metalloprotein that protects the mitochondria from ROS damage [55].

Uncoupling proteins (UCPs), which comprise a set of five similar proteins, are situated in the inner mitochondrial membranes of various tissues. They play a role in a variety of functions and cellular activities, ranging from regulating the body temperature to influencing insulin release and providing neuroprotection [56]. Amongst all tissues, skeletal muscle possesses the widest array of UCPs, with all five types present in its mitochondria. As a result, skeletal muscle has been extensively researched to enhance our understanding of UCP functions and related disorders. Over the past thirty years, UCPs have been extensively examined due to their impact on glucose and lipid metabolism. Furthermore, numerous studies that involved mice, rats and humans demonstrated the significant protective effects of mitochondrial UCPs against oxidative stress and issues with mitochondrial function. Nevertheless, their precise role remains incompletely understood. Recent research underscores the critical roles of mitochondrial proteins, such as UCP1/3 and mitochondrial protein adenine nucleotide translocase 1 (ANT1), in the pathogenesis of DbCM [57].

UCP1–UCP3 are involved in the regulation of mitochondrial proton leakage and the dissipation of the proton gradient, which can modulate reactive ROS production and protect against oxidative stress in cardiac cells [58]. On the other hand, ANT1, which is a key component of the mitochondrial ADP/ATP translocase system, plays a pivotal role in maintaining energy homeostasis by facilitating the exchange of ADP and ATP across the mitochondrial inner membrane [57].

The dysregulation of ANT1 can lead to impaired ATP production and energy deficits, which contributes to the contractile dysfunction characteristic of a diabetic heart [59]. Exploring the interplay between UCP1-3 and ANT1 could reveal novel insights into their potential as therapeutic targets to ameliorate mitochondrial dysfunction and improve cardiac outcomes in diabetic patients [60].

## 4. Molecular Mechanisms Determining Cardiac Remodeling

Cardiac hypertrophy and fibrosis are the main features of DbCM determined by different mechanisms [61,62]. Increased insulin levels with the consequent higher levels of insulin-like growth factor (IGF-1) activate the mitogen activated protein kinase 1 (Erk1/2) and phosphoinositide 3-kinases (PI3K) signaling pathways, which, in turn, induce cardiomyocyte hypertrophy [63]. The mammalian target of rapamycin complex 2 (mTORC2) signaling pathway functions as an effector of insulin/IGF-1; it activates and phosphorylates protein kinase B (PKB), which modulates key substrates, including the transcription factors of FoxO1/3a, the metabolic regulator (GSK3b) and the mTORC1 inhibitor (TSC2) [64], resulting in cell production and proliferation. Hyperglycemia exerts its deleterious effects mainly through chronic hexosamine biosynthetic pathway (HBP) activation; this leads to excessive O-GlcNAcylation of target proteins, which, in turn, can contribute to increased oxidative stress, impairment of DNA repair machinery, enhanced apoptotic and autophagic stimuli and mitochondrial dysfunction [65,66,67,68]. This all contributes to cardiac fibrosis, hypertrophy and increased susceptibility to ischemic injury [69,70,71,72,73]. Oxidative stress contributes to cardiac hypertrophy via the increased expression of nitrogen oxide (NOX) proteins due to the dysregulation of PPARα and angiotensin II [74,75]. In animal models, NOX 2 and NOX4 are associated with cardiac hypertrophy and fibrosis [76]. In diabetic cardiac tissue, collagen deposition in the extracellular matrix (ECM) is enhanced, mainly due to the increased deposition of collagen types I and III. The activation of Transforming Growth Factor-b1 (TGF-b1) and wingless-related integration site (WNT) pathways, together with an accelerated extracellular matrix degradation due to the remodeling of matrix metalloproteinases (MMPs) and decreased availability of nitric oxide (NO), lead to a dysregulated process, with the consequence of high cardiac collagen deposition and consequent interstitial fibrosis [77,78]. MMP release is enhanced by inflammation [79]. The activation of these processes is the consequence of the stimulation of increased renin–angiotensin–aldosterone (RAAS) activity, increased AGEs, insulin resistance and hyperglycemia [80,81,82]. AGEs accumulate in different cell types, which affects the extracellular and intracellular structure, causing crosslinks in collagen molecules, thereby making it more difficult to be degraded and stimulating cardiac fibroblasts. This, in turn, leads to increased fibrosis with consequent increased myocardial stiffness [83].

Hyperglycemia is also a direct contributor to cardiac fibrosis. In experimental models, cardiac fibroblasts exposed to high glucose synthesize excessive amounts of collagen and other ECM proteins [83]. As discussed before, other factors such as adipokines release, oxidation of FAs as a source of energy and neurohumoral pathways can activate fibroblasts and, consequently, enhance myocardial fibrosis [84]. Fibroblast activation is also triggered by the accumulation of harmful lipids in cardiomyocytes and successive activation of inflammatory and fibrogenic programs [85]. In DbCM, higher rates of apoptosis and autophagy of cardiomyocytes, fibroblasts and endothelial cells were observed, which were caused by hyperglycemia, insulin resistance, lipid peroxidation, increased RAAS activation, oxidative stress and endoplasmic stress [86]. Particularly, hyperglycemia induces cell death through the local increase of angiotensin II and by inducing lysosomal membrane damage and enhanced cathepsin D expression and lysosomal release [87,88]. Furthermore, in T2DM, an alteration of Treg/Th17 and Treg/th1 is documented and leads to a pro-inflammatory subset that contributes to cardiac cell death [89,90]. Cardiomyocyte apoptosis stimulates the deposition of collagen through cardiac fibroblast activation and proliferation and cardiac interstitial fibrosis [91]. Heart stiffness and hypertrophy causes decreased complacency and the subsequent development of systolic and diastolic dysfunction [92].

## 5. Myocardial Calcium Handling

Myocyte contraction needs an appropriate calcium balance, as the binding of calcium to troponin C in the sarcomere is responsible for the conformational changes that lead to heart contraction [93]. In DbCM, perturbations in calcium handling have been described [94]. L-type calcium channels (LTCCs) are voltage-gated channels located on the sarcolemma activated by the influx of sodium ions caused by the arrival of action potential at the cardiac myocytes.

The activation of LTCCs causes an influx of calcium ions. In animal models of diabetic mice with heart impairment, a decrease in the expression of LTCC was observed, with a consequent reduction in the calcium current [95].

Also, the exposure of cardiomyocyte to AGEs causes increased ROS and consequent calcium transients reduction [96]. In cardiac myocytes produced from induced pluripotent stem cells (iPSC-CMs) and exposed to a diabetic environment, calcium transient frequency and amplitude reduction was observed [97].

However, in animal models, an increase in the amplitude of calcium transients in myocytes was documented, and this effect has been considered to be caused by a dysfunction in the sodium/calcium exchanger, independently of LTCCs [98]. Furthermore, patients with T2DM have an increased expression of SLC8A1, which is a gene that encodes the sodium/calcium exchanger [99].

Calcium binds to ryanodine receptor-2 (RyR2) on the sarcoplasmic reticulum (SR), which leads to an extensive release of calcium from the SR into the cytosol; this, in turn, allows calcium to bind to troponin C with subsequent destabilization of the troponin–tropomyosin complex from the actin–myosin binding site, causing cross-bridges formation and producing tension; AGEs can contribute to impaired Ryanodine receptor (RyR2) [100,101].

SR calcium ATPase (SERCA-2) is a channel responsible for the reuptake of calcium back into the SR; in diabetic mice, a reduced expression of SERCA-2 [94,102] and an AGE decrease has led to SERCA-2 activity being observed [103].

Overall, cytosolic calcium overload can be observed in cardiomyocytes of T2DM individuals, particularly in end-stage heart failure (Table 1; Figure 2).

## 6. Epigenetic Changes

Several studies indicate that elevated glucose levels likely play a significant role in the epigenetic regulation of numerous genes, which influences their expression. Consequently, epigenetic alterations could offer a biological rationale for the complexities of DbCM [104,105]. In addition, some gene variants can increase the risk of hypertension in obese children and adolescents, who are the subjects that are mainly predisposed to develop T2DM [104,105].

The main epigenetic changes represented by DNA methylation; alteration of the expression of microRNAs (miRNAs), circular RNA (circRNAs) and long non-coding RNAs (lncRNAs); and alterations to histones (acetylation and methylation) are significantly involved in the development of DbCM [106,107].

### 6.1. DNA Methylation/Demethylation

Hyperglycemia and the body’s reduced response to insulin (insulin resistance) are closely linked to DNA methylation. DNA methylation controls the activity of genes connected to DbCM, leading to a decline in the heart’s ability to contract, an increase in oxidative stress, cardiac restructuring, cell death in heart muscle cells (cardiomyocytes), and the initiation and advancement of DbCM. Hence, regulating DNA methylation to counteract impaired insulin sensitivity could offer an encouraging avenue for the quick management and restoration of sugar levels, potentially halting the progression of DbCM [108].

DNA methylation involves the adding of 5-methylcytosine through DNA methyltransferases (DNMTs) [109]. Irregular DNA methylation is engaged in conditions like diabetes and cardiovascular diseases, hypertension, coronary artery atherosclerosis and cardiac failure [110]. There is a positive correlation between oxidative stress and cardiac insulin resistance. A study indicated that demethylation of the Nuclear factor erythroid erythtroid 2-related factor 2 (Nrf2) promoter reduces Nrf2 activity, boosts Keap1 protein expression, hampers the transcription of various antioxidant defense genes, causes oxidative damage and worsens insulin sensitivity [111]. Furthermore, in hyperglycemic conditions, irreversible changes in DNMT activity can occur. The activation of the pro-inflammatory cytokine TNF-ɑ might raise the levels of DNA methyltransferase, which leads to the methylation of the promoter region of the SERCA2a. This results in the decrease in SERCA2a levels, which causes an overload of calcium ions and contributes to dysfunction in the relaxation phase of the heart (diastolic heart failure) and the progression of cardiac failure in the most serious cases. In conditions characterized by high blood sugar levels and insulin resistance, the RAAS is triggered. This activation triggers the methylation of the promoter of the angiotensin type II-1b (AT1b) gene, which leads to the increased expression of AT1b and the promotion of enlargement of the heart muscle (myocardial hypertrophy) [112,113].

### 6.2. Histone Modifications

Histone methylation, mainly on lysine and arginine residues, is regulated by enzymes that ensure orderly gene expression.

The modification of histones through processes such as methylation, acetylation and phosphorylation affects the transcriptional activity of genes, which results in changes in gene expression [114]. Histone acetylation, which is facilitated by histone acetyltransferases (HATs) and histone deacetylases (HDACs), are crucial for gene regulation and maintaining genomic wholeness. HDACs are known to play a significant role in DbCM, such as cardiac fibrosis and hypertrophy [115]. Different HDACs have different effects on DbCM; the SIRT family, especially SIRT1, is involved in controlling factors related to metabolism, adipogenesis and insulin secretion. The interplay between HATs and HDACs impacts the NF-κB transcriptional activity, thereby molding the expression of inflammatory genes downstream. When exposed to heightened glucose levels, cultured monocytes show an augmented recruitment of HATs CPB and p/CAF, leading to heightened histone lysine acetylation at promoters of inflammatory genes. The elevation in histone acetylation levels in monocytes from T2D patients compared with healthy underscores its role in inflammation. Beyond glucose, oxidized lipids are implicated in the enhancement of inflammatory gene promoter histone acetylation in a promoter histone acetylation in a CREB/p300 (HAT)-dependent manner, consequently amplifying gene expression [116]. The protein p300 proves influential in pivotal signaling pathways responding to oxidative stress induced by elevated glucose, which influences extracellular matrix (ECM) components within diabetic-affected organs.

Elevations in p300 levels due to high glucose concentrations lead to heightened histone acetylation at promoters of crucial ECM genes and vasoactive factors in endothelial cells [117]. Curcumin acting on the p300 inhibitor aided in curtailing aberrant gene expression changes triggered by hyperglycemia, especially those linked to diabetic complications and cardiomyocyte hypertrophy. These findings collectively underscore the pivotal role played by chromatin histone acetylation in dictating gene expression patterns associated with complications arising from diabetes.

### 6.3. Non-Coding RNAs

MicroRNAs, or miRNAs, are short, non-coding RNA molecules that play a crucial role in regulating gene expression within cells. As seen in previous studies, specific miRNA profiles can represent specific biomarkers for detecting an increased risk of developing metabolic dysfunction in obese children [118]. In diabetic heart tissues, approximately 30% of miRNA showed altered expression patterns [119]. These miRNAs are implicated in the modulation of cardiac fibrosis and hypertrophy in DbCM, and the processes oxidative stress, apoptosis and cardiac remodeling [120].

MicroRNAs are involved in either promoting or inhibiting apoptosis in cardiomyocytes. miR-203 can alleviate oxidative damage and the fibrosis of cardiac cells in DbCM by targeting PIK3CA to inhibit the PI3K/Akt signaling pathway [121]. An increased expression of miR-22 in DbCM was shown to help counteract oxidative stress [122]. The inhibition of miR-144 reduces oxidative stress, decreases STZ-induced (streptozotocin) cardiomyocyte apoptosis and enhances heart function in DbCM mice through the Nrf2 pathway [123]. The knockdown of miR-451 impaired the transition from endothelial to mesenchymal states, reduced heart fibrosis and enhanced cardiac function in diabetic mice [124]. Inversely, the amplification of miR-200b also produces protective actions. Transforming growth factor beta (TGF-β) plays a key role in the process of endothelial-to-mesenchymal transition (EMT). miR-142-3p can hinder EMT induced by high blood sugar levels and prevent the progression of diabetic cardiomyopathy to heart failure by acting on the TGF-β/Smad pathway. A reduced miR-133a expression in diabetic mouse cardiomyocytes under high glucose conditions inhibits myocardial hypertrophy when miR-133a is overexpressed [125]. Furthermore, blocking miR-199a activates PGC-1α, which enhances mitochondrial fatty acid oxidation and reverses myocardial hypertrophy [126]. A decreased miR-1 expression in cardiomyocytes under high glucose conditions can be mitigated by boosting miR-1 levels to prevent diabetes-induced oxidative stress in the cardiomyocytes. For instance, miR-1 is able to suppress IGF-1 expression, trigger cytochrome c release and enhance apoptosis in cardiomyocytes [127]. Hsp60 plays a crucial role in the protection against DbCM, and studies showed that miR-1 and miR-206 can increase cardiomyocyte apoptosis under high-glucose conditions by inhibiting Hsp60 expression [128]. An increased expression of miR-532 in the heart tissue of T2DM individuals reduces the expression of the anti-apoptotic protein ARC [129]. miR-150-5p is involved in regulating lymphocyte development and inflammatory cytokine production. The inhibition of miR-150-5p was shown to improve cardiac inflammation and fibrosis by targeting Smad7 in heart fibroblasts exposed to high glucose [130]. miRNA associated with an adverse profile includes the elevated expression of the miR-212/132 family in diabetic rat cardiomyocytes, which activates the transcription factor FoxO3 and results in myocardial hypertrophy and autophagy. miR-451 may worsen cardiac muscle cell lipotoxicity and high-fat diet-induced lipotoxicity, which leads to cardiac hypertrophy in DbCM mice by blocking the hepatic kinase B1/AMP-activated protein kinase (LKB1/AMPK) pathway [131]. miRNAs and ROS have a complex interaction in DbCM; for instance, miR-210 may influence mitochondrial damage that is closely related to oxidative stress by targeting ROS-related molecules [132]. miR-30c and miR181a can collaboratively enhance p53 and p21 gene expression, which leads to cardiac cell hypertrophy and apoptosis in rats with diabetic cardiomyopathy (DbCM) and cardiomyocytes treated with high blood sugar. In the hearts of type 2 diabetic mice lacking the leptin receptor and treated with STZ, miR-195 is significantly upregulated, which results in the downregulation of its targets SIRT1 and Bcl2, ultimately causing cardiomyocyte apoptosis [133]. Additionally, miRNAs, like miR-700, miR-223 and miR-146, also play a role in the onset of DbCM through modulating inflammatory responses, although further research is necessary in this area.

**Table 1 ijms-25-08280-t001:** Principal studies that analyzed molecular mechanisms in diabetic cardiomyopathy.

Main Mechanism	Ref.	In Vitro/In Vivo/Ex Vivo	Molecular Evidence
Oxidative stress	[20]	In vivo, STZ-induced diabetic rats	↑ ROS, TNF-α, RAGE and NFkB-p65
Inflammation, oxidative stress, cardiac hypertrophy and fibrosis	[23]	In vivo, STZ-induced diabetic mice In vitro, AC16 human myocardial cells treated with high glucose	In STZ-induced diabetic mice: ↑ cardiomyocyte area ↑ collagen deposition ↑ IL-1β, IL-6, IL-18 and TNFα ↑ activation of NLRP3 In AC16 cells: ↑ phosphorylation levels of NF-κB p65 and IκB ↑ ROS
Inflammation, oxidative stress, apoptosis, pyroptosis, cardiac remodeling and ventricular dysfunction	[24]	In vivo, STZ-induced diabetic mice In vitro, high-glucose-treated H9c2 cardiomyocytes	↑ NLRP3, ASC, caspase-1 and IL-1β NLRP3 gene silencing reduces left ventricular dysfunction in DbCM and reversed myocardial remodeling In H9c2 cells: ↑ ROS and phosphorylation of NF-kB p65 In cells pretreated with NAC: ↓ ROS activity, NF-kB, NLRP3, ASC, pro-caspaspe-1, activated caspase-1, pro- IL-1β and mature IL-1β
Inflammation and pyroptosis	[31]	In vitro, H9c2 rat cardiomyoblast cell line treated with high glucose	↑ LDH, IL-1β, IL-18, NLRP3, GSDMD-N, caspase-1 and IL-1β ↑ miR-223-3p
Inflammation and mitochondrial dysfunction	[32]	In vivo, mice overexpressing IGFII in pancreatic beta cells in hyperlipidemic background with T2DM fed with high-fat diet for 12 weeks with induced MI	Impaired mitophagy in peri-infarct regions of LV, ↑ mtDNA, activation of Aim2, NLRC4 inflammasome, caspase-I, IL-18 and cardiomyocyte death
Interstitial fibrosis	[33]	In vivo, STZ-induced diabetic mice	LV systolic functional impairment and ↑ interstitial fibrosis
Inflammation	[42]	In vivo, STZ-induced diabetic mice	Release of mtDNA and activation of cGAS-STING signaling pathway leading to the activation of NLRP3 and ↑ TNF-α, IFN-β, IL-1β and IL-18
Mitochondrial dysfunction	[45]	In vivo, T2DM human patients	↓ myocardial glucose utilization rate
Mitochondrial dysfunction	[48]	In vivo, STZ-induced diabetic mice	↑ PPAR alpha, ACO and M-CPTI
Mitochondrial dysfunction	[53]	In vivo, STZ-induced diabetic mice In vitro, HL-1 murine cardiomyocytes	In STZ-induced diabetic mice: ↓ HIF-1α after ischemia In HL-1 cardiomyocytes: failure of hypoxia-mediated metabolic adaptation with increased lactate efflux and glucose consumption
Mitochondrial dysfunction and oxidative stress	[55]	In vivo, STZ-induced diabetic mice	↑ ROS, catalase activity, SOD, TFAM, PGC-1α and CO1 ↓ electron transport chain complexes and SIRT3 activity ↑ acetylation of MnSOD
Mitochondrial dysfunction	[67]	In vivo, T1D mice	↑ 8-OHdG, O-GlcNAcylation of Ogg1 activity and mtDNA damage
Mitochondrial dysfunction	[68]	In vivo, STZ-induced diabetic mice	↑ mitochondrial OGT, ↓ mito-specific O-GlcNAcase (OGA), OGT is mislocalized and ↓ interaction of OGT and complex IV
Cardiac hypertrophy	[69]	Ex vivo, cardiomyocytes isolated from type 2 diabetic db/db mice	↑ HBP and *O*-GlcNAc levels
Cardiac hypertrophy	[70]	In vitro, endothelial vein cells of T2DM patients	↑ O-GlcNAc levels
Cardiac dysfunction	[73]	In vivo, polygenic T2DM mice In vitro, isolated coronary endothelial cells from mice	↑ coronary endothelial cells apoptosis, ↓ coronary flow velocity reserve, ↓ cardiac contractility and ↑ p53 protein levels
Cardiac fibrosis and inflammation	[77]	In vivo, STZ-induced diabetic mice	LV dysfunction, diastolic stiffness; ↑ TGFβ, IL1β, and fibrosis; ↓ MMP-2 activity
RAS, apoptosis and cardiac fibrosis	[80]	In vivo, STZ-induced diabetic mice	↑ intracellular ATII, angiotensinogen and renin in cardiac myocytes Superoxide production, myocyte apoptosis and cardiac fibrosis were inhibited by RAS inhibitors
Cardiac fibrosis	[82]	In vitro, differentiated murine mesangial cells exposed to D-glucose	↑ incorporation of 3[H]proline ↑ stimulation of collagen types I and IV by high D-glucose
RAS and cardiac fibrosis	[83]	In vitro, neonatal rat ventricular fibroblasts exposed to high glucose	↑ intracellular ATII and TGFβ, ↑ collagen-1 synthesis by cardiac fibroblasts that is inhibited by RAS inhibitors
Myocyte death and cardiac hypertrophy	[87]	In vivo, STZ-induced diabetic mice	↑ myocyte apoptosis, AT, AT II, renin and AT1
Myocyte death	[88]	In vitro, neonatal rat ventricular cardiomyocytes cultured with glucose	↓ number of lysosomes with acidic pH ↑ Galectin3-RFP puncta and leakage of CTSD
Inflammation and myocyte death	[89]	In vivo, T2DM patients	↓ ratios of CD4(+)CD25(hi) Treg/Th17 cells and CD4(+)CD25(hi) Treg/Th1 cells
Alterations of myocardial calcium handling	[95]	In vivo, sedentary db/db mice	↑ diastolic SR-Ca(2+) leak ↓ synchrony of Ca(2+) release, transverse-tubule density, peak systolic and diastolic Ca(2+), caffeine-induced Ca(2+) release and SR Ca(2+) ATPase-mediated Ca(2+) uptake during diastole rate
Alterations of myocardial calcium handling and oxidative stress	[96]	In vitro, neonatal rat cardiomyocytes treated with AGEs	↓ calcium transient amplitude and sarcoplasmic reticulum calcium content ↑ ROS, NADPH oxidase activity, activation of p38 kinase and nuclear translocation of NF-κB with the subsequent induction of inducible NO synthase expression
Hypertrophy, apoptosis and alterations of myocardial calcium handling	[97]	In vitro, pluripotent stem cells generated from urine epithelial cells of T2DM patients	Irregular Ca(2+) transient waveforms, decreased transient amplitude, shorter transient duration, shorter decay, slower maximal rising rate and slower maximal decay rate ↑ apoptosis, cellular hypertrophy and lipid accumulation
Alterations of myocardial calcium handling	[98]	In vivo, STZ-induced diabetic mice	↑ Ca(2+) transient ↓ Na(+)/Ca(2+) exchanger current
Alterations of myocardial calcium handling	[99]	In vivo, T2DM patients	↓ left ventricular ERG, KCNH2 and KCNJ3 gene expression ↑ NCX1, KCNJ2, KCNJ5 and SLC8A1 gene expression ↑ QT interval
AGEs and alterations of myocardial calcium handling	[101]	In vivo, STZ-induced diabetic mice	AGEs formation on intracellular RyR2
Alterations of myocardial calcium handling	[102]	In vitro, HEK-293 cells	↑ Ca(2+) uptake velocity for expressed SERCA2 by exposure to CaM kinase I1
AGEs and alterations of myocardial calcium handling	[103]	In vivo, STZ-induced diabetic mice	↓ heart relaxation ↓ SERCA2a expression ↑ PLB
Epigenetic changes	[107]	In vivo, STZ-induced diabetic mice In vivo, formalin-fixed and paraffin-embedded human myocardial archived tissues In vitro, neonatal rat cardiomyocytes exposed to glucose	Absence of methylation in the promoter regions of the DUSP-1
Epigenetic changes	[111]	In vivo, nondiabetic women	Positive association between the methylation levels of the CpG site 783 with insulin sensitivity
Epigenetic changes	[112]	In vitro, HL-1 cardiomyocytes	↓ SERCA2a RNA and protein expression ↑ methylation in the SERCA2a promoter region ↑ DNA methyltransferase 1 expression
Epigenetic changes	[116]	In vitro, vascular smooth muscle cells	12(S)-HETE activated Src, focal adhesion kinase, Akt, p38MAPK, CREB, expression of monocyte chemoattractant protein-1, IL-6 genes and histone H3-Lys-9/14 acetylation on their promoters
Epigenetic changes	[117]	In vitro, neonatal rat cardiomyocytes exposed to glucose In vivo, STZ-induced diabetic rats	Cellular hypertrophy and ↑ mRNA expression of ANP, BNP, ANG, MEF2A, MEF2C and transcriptional coactivator p300 In STZ-induced diabetic rats: ↑ ANP, BNP, ANG, mRNA, p300, MEF2A, and MEF2C expression
Epigenetic changes	[119]	In vivo, STZ-induced diabetic mice	Dysregulation of 316 out of 1008 total miRNAs implicated in myocardial signalling networks that trigger apoptosis, fibrosis, hypertrophic growth, autophagy oxidative stress and heart failure
Epigenetic changes	[121]	In vivo, STZ-induced diabetic mice	Upregulation of miR-203 inhibited the activation of PI3K/Akt signaling pathway and ↓ PIK3CA, PI3K, Akt, CoI I, CoI III, ANP, MDA and ROS in the myocardial tissues
Epigenetic changes	[123]	In vivo and ex vivo, STZ-induced diabetic mice In vitro, cultured cardiomyocytes	In STZ-induced diabetic mice: ↓ miR-144 in heart tissues In cultured cardiomyocytes: high glucose exposure induced ↓ miR-144
Epigenetic changes	[124]	In vivo, STZ-induced diabetic mice	miR-451 knockdown attenuated cardiac fibrosis, improved cardiac function and suppressed endothelial-to-mesenchymal transition
Epigenetic changes	[126]	In vivo, mice treated with anti-miR-199a	Upregulation of genes related to cytoplasmic translation and mitochondrial respiratory chain complex assembly
Epigenetic changes	[127]	In vitro, H9C2 cells exposed to high glucose	↑ miR-1 expression level ↓ mitochondrial membrane potential ↑ cytochrome-c release and apoptosis
Epigenetic changes	[128]	In vivo, STZ-induced diabetic mice In vitro, cardiac muscle cell line HL-1	In STZ-induced diabetic mice: ↓ GAS5 ↑ NLRP3, caspase-1, Pro-caspase-1, IL-1β and IL-18 GAS5 overexpression improved cardiac function and myocardial hypertrophy In cardiac muscle cell line HL-1: ↑ NLRP3, caspase-1, Pro-caspase-1, IL-1β and IL-18,
Epigenetic changes	[129]	In vivo, STZ-induced diabetic mice In vitro, H9C2 cells exposed to high glucose	STZ-induced diabetic mice: ↑ p53 and p21 ↓ miR-30c and miR-181a H9C2 cells exposed to high glucose: ↑ p53 and p21 ↓ miR-181a
Epigenetic changes	[130]	In vivo, STZ-induced diabetic mice	↑ miR-195 expression
Epigenetic changes	[131]	In vivo, mice with obesity and diabetes induced by high-fat diet In vitro, neonatal rat cardiac myocytes stimulated with palmitic acid	In neonatal rat cardiac myocytes: ↑ miR-451 expression in a dose- and time-dependent manner In cardiomyocyte-specific miR-451 knockout mice: cardiac hypertrophy and contractile reserves were ameliorated, ↑ Cab39 and phosphorylated AMPK, and ↓ mTOR
Epigenetic changes	[133]	In vivo, T2DM human patients undergoing coronary artery bypass graft surgery In vivo, type-2 diabetic mice In vitro, human ventricular cardiomyocytes treated with high glucose	↑ miR-532 expression in the right atrial appendage tissue, which was associated with downregulation of ARC In human ventricular cardiomyocytes (AC16) treated with high glucose: inhibition of miR-532 activity in high-glucose-cultured human cardiomyocytes prevented the downregulation of ARC and attenuated apoptotic cell death

↑: increased; ↓: decreased. ACO, 1-Aminocyclopropane-1-Carboxylic Acid Oxidase; AIM2, absent in melanoma 2; ANG, Angiogenin; ANP, Atrial Natriuretic Peptide; ARC, apoptosis repressor with caspase recruitment domain; ASC, apoptosis associated speck like protein; AT, angiotensin; ATP, Adenosine triphosphate; BNP, Natriuretic Peptide B; Ca, calcium; CaMKII, Ca(2+)/calmodulin-dependent protein kinase II; CO1, cytochrome oxydase I; CPT1B, carnitine palmitoyltransferase 1B; CREB, Cyclic adenosine monophosphate response element-binding protein; CTSD, cathepsin D; DbCM, diabetic cardiomyopathy; DUSP1, Dual Specificity Phosphatase 1; FDG, fluorine-18 fluorodeoxyglucose; GSDMD, Gasdermin D; HBP, hexosamine biosynthesis pathway; HIF, hypoxia inducible factor; IFN, interferon; IGF, insulin-like growth factor; Il, interleukin; KCNJ, potassium inwardly rectifying channel subfamily J; LDH, lactate dehydrogenase; LV, left ventricle; LVFS, left-ventricular fractional shortening; MAPK, mitogen-activated protein kinases; membrane carnitine palmitoyltransferase, M-CPT I; MEF, Myocyte Enhancer Factor; MI, myocardial infarction; miR, microRNA; MnSOD, Manganese Superoxide Dismutase; mtDNA, mitochondrial DNA; mTOR, phosphorylated mammalian target of rapamycin; Na, sodium; NAC, N-acetylcysteine; NADPH, Nicotinamide adenine dinucleotide phosphate; NCX, Na(+)-Ca(2+) exchanger; NFkB, nuclear factor-kB; NLRC4, NLR Family CARD Domain Containing 4; NLRP3, NOD-like receptor family pyrin domain containing 3; NO, nitric oxide; Nrf2, nuclear factor-erythroid 2-related factor 2; OGT, *O*-GlcNAc transferase; PET, Positron Emission Tomography; PGC1alpha, Peroxisome proliferator-activated receptor-gamma coactivator 1 alpha; PIK3, Phosphoinositide 3-kinase; PLB, Phospholamban; PPAR, peroxisome proliferator-activated receptor; PPARGC1A, PPAR coactivator 1-alpha; PP2, Protein phosphatase 2; RAGE, Advanced glycated end products receptor; RAS, renin angiotensin system; ROS, reactive oxygen species; SERCA, Sarco-Endoplasmic Reticulum Calcium ATPase; SLC8A1, Solute Carrier Family 8 Member A 1; SIRT3, NAD-dependent deacetylase sirtuin-3; SOD, Superoxide Dismutase; SR, sarcoplasmic reticulum; STING, stimulator of interferon genes; STZ, streptozotocin; TFAM, Mitochondrial transcription factor A; TGF, Transforming growth factor; TNF-α, Tumor Necrosis factor alpha; T1D, type 1 diabetes; T2DM, type 2 diabetes mellitus; 8-OHdG, 8-hydroxy-2′-deoxyguanosine.

## 7. Prospective Treatments

To date, there is not a specific therapeutic approach for DbCM in clinical practice; the increasing knowledge on molecular mechanisms of DbCM may open new possibilities for developing therapies that target specific pathways (Table 2). Some drugs already used for the treatment of T2DM were studied to explore their potential role on DbCM, such as Glucagon-like Peptide-1 (GLP-1). Receptor agonists ameliorate cardiomyocytes’ calcium handling and have anti-inflammatory effects [134,135]. Metformin, in addition to its classical actions, can improve mitochondrial biogenesis and enhance antioxidant activity, decrease mitochondrial ROS production, upregulate mitophagy and normalize calcium handling [136,137,138,139]. Clinical studies are also exploring the potential role on DbCM of drugs used for other diseases. Cyclic guanosine monophosphate-phosphodiesterase type 5 (PDE5) inhibitors, fenofibrate and alpha-lipoic acid can inhibit NLRP3 inflammasome-mediated pyroptosis [140]. Levosimendan and istaroxime may have a role in improving calcium handling in cardiomyocytes by stabilizing the calcium-troponin C complex and increasing the expression of SERCA-2 and sodium-calcium exchanger-1, respectively [141,142]. The inhibition of HDACs can attenuate cardiac hypertrophy and fibrosis and can improve cardiac function and reverse cardiac remodeling [143,144]. The most promising medications in treating of DbCM and cardiovascular disease in T2DM patients are sodium-glucose cotransporter-2 inhibitors (SGLT-2is), also known as gliflozins. SGLT2is drastically reduce the risk of cardiovascular death patients with heart failure according to a recent series of large clinical trials, including the Dapagliflozin and Prevention of Adverse Outcomes in Heart Failure (DAPA-HF), the Empagliflozin Outcome Trial in Patients with Chronic Heart Failure and a Reduced Ejection Fraction (EMPEROR-Reduced), and the Empagliflozin Outcome Trial in Patients with Chronic Heart Failure with Preserved Ejection Fraction (EMPEROR-Preserved) [145,146,147,148,149]. SGLT-2is have antioxidant and anti-inflammatory properties, can mitigate cardiac fibrosis and have a positive impact on calcium handling [140,150,151,152,153]. SGLT-2is prevent glucose reabsorption in the proximal tubule kidneys, increasing urinary glucose excretion and lowering plasma glucose. Through a variety of tissue- and organ-specific anti-inflammatory effects, SGLT2is may be able to provide cardiovascular protection. For instance, hyperuricemia, which is frequently observed in people with heart disease [154], seems to support the pathophysiology of cardiovascular disease by promoting the proliferation of vascular smooth muscle cells and raising the synthesis of pro-oxidative [155]. Given its demonstrated ability to lower uric acid levels, SGLT2is may have the potential to improve this diseased route. By increasing the uric acid excretion in urine, SGLT2is lower uric acid levels. Another proposed mechanism involves epicardial adipose tissue. Epicardial adipose tissue secretes high levels of activin A, which is a cytokine that enhances inflammation, in people with DbCM [156]. It was demonstrated that dapagliflozin and canagliflozin reduce epicardial adipose tissue in people with type 2 diabetes (T2D), which lowers vascular inflammation and fibrosis. SGLT2is significantly decreased the left ventricular mass in patients with DbCM according to a recent meta-analysis of a double-blinded, placebo-controlled, randomized trial that evaluated left-ventricular remodeling by cardiac magnetic resonance imaging [157]. Moreover, studies that employed murine models of myocytes demonstrated that dapagliflozin and empagliflozin can prevent cardiac fibrosis [158,159]. Similarly, according to preliminary data on human myofibroblasts, empagliflozin may directly influence extracellular matrix reorganization through cell-mediated collagen remodeling and the reduction of myofibroblast activity [160]. In T2DM, there is a slight and sustained increase in blood ketones. In this condition, the heart uses more beta-hydroxybutyrate—which is a ketone—than usual to produce energy instead of fatty acids [161]. It enhances the mitochondrial level conversion of oxygen consumption into work efficiency [162]. It was proposed that SGLT2is cause the metabolism to shift toward the oxidation of FAs and promote ketogenesis based on research with dapagliflozin and empagliflozin. Because of this change, beta-hydroxybutyrate plasma levels are higher. This may provide substantial cardioprotection for T2DM patients [152,163]. Moreover, hemoconcentration induced by SGLT2is increases oxygen release. Thus, such a metabolic substrate change can provide cardioprotection [164]. In animal models, the suppression of overexpressed miRNA in heart failure was studied with promising results [165]. Specific miRNA and lncRNA profiles could also be potentially used as biomarkers for early detection of DbCM, or to identify diabetic individuals with an increased risk of developing cardiac dysfunction. It was hypothesized that miR-9, miR-21, miR-29, miR-30d, miR-34a, miR144, miR150, miR-320 and miR378 in particular could be associated with a higher risk of cardiomyopathy [120]. Lastly, a complete knowledge of molecular mechanisms of DbCM could help to better understand the different phenotypes of early-onset DMT2 and the pathological basis of the worst cardiac complications, with the aim of developing tailored treatments.

## 8. Conclusions

Patients with T2DM require multidisciplinary surveillance, including cardiac follow-up, to exclude the development of DbCM. The pathogenetic basis of DbCM involves multiple molecular pathways and metabolic changes. Knowing these mechanisms helps to establish effective therapies that may prevent the development of DbCM or treat overt disease conditions. Providing that lifestyle changes and adequate diet should always be pursued to aim for optimal glycemic control, several therapeutic options may be employed. Among them, SGLT-2is already represent a proven effective therapeutic option in improving the outcome of T2DM patients with heart disease. By acting on multiple pathogenic pathways, we can assist with reducing the mortality and morbidity due to DbCM.

## Figures and Tables

**Figure 1 ijms-25-08280-f001:**
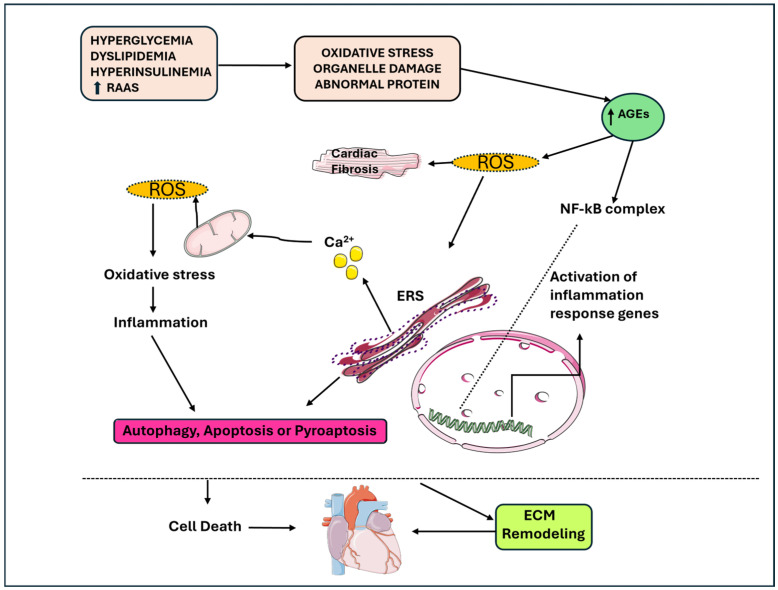
The link between apoptosis, autophagy, pyro-apoptosis, and cardiomyocyte loss in DbCM, inflammation and metabolic stress contribute to the senescence of cardiomyocytes. AGEs: advanced glycation end products; DbCM: diabetic cardiomyopathy; ERS: stress endoplasmic reticulum; ROS: reactive oxygen species.

**Figure 2 ijms-25-08280-f002:**
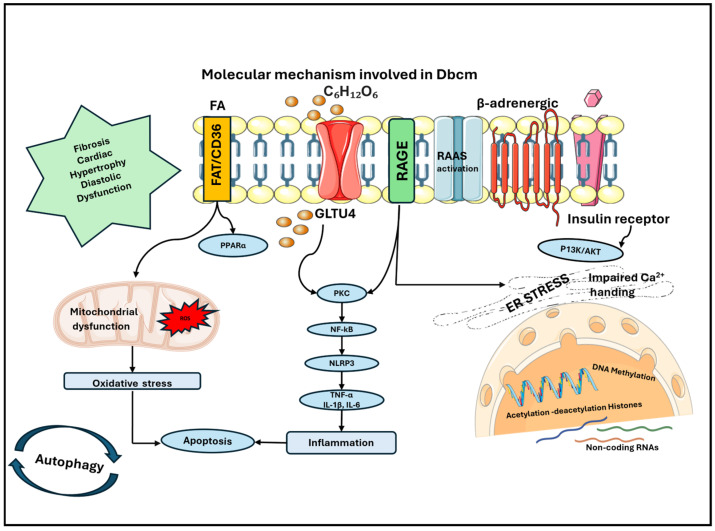
Molecular mechanisms involved in the development of DbCM: accumulation of FAs contributes to mitochondrial dysfunction with increased production of ROS and consequent oxidative stress in cardiomyocytes. This causes a higher rate of apoptosis, which is also enhanced by inflammation that is sustained by the production of cytokines. Cytokines are released through the activation of NF-kB, which is mainly sustained by hyperglycemia, the accumulation of AGEs and the activation of their receptor AGE. Other contributing mechanisms are the disarrangement of RAAS, the stress on ER and epigenetics, which is induced by hyperglycemia and hyperinsulinism, which contribute to modification of the gene expression and cellular phenotype. The principal perturbations in the heart are fibrosis, cardiac hypertrophy and diastolic dysfunction, which are all main features of DbCM. AGE: advanced glycation end product; DbCM: diabetic cardiomyopathy; ER: endoplasmic reticulum; FAs: fatty acids; RAAS: renin angiotensin aldosterone system; ROS: reactive oxygen species.

**Table 2 ijms-25-08280-t002:** Drugs with potential effects on DbCM.

	Primary Effect	Primary Indication	Pleiotropic Effect in DdCM
GLP-1 receptor agonists	Inhibits glucagon release	T2DM	Ameliorates calcium handling
	Stimulates insulin incretion		Anti-inflammatory effect
	Slows gastric emptying		Reduces oxidative stress
Metformin (AMPK activator)	Inhibits the RC in the liver	T2DM	Enhances antioxidant activity
	Lowers glucose production		Decreases mitochondrial ROS production
	Increases insulin sensitivity		Upregulates mitophagy
			Normalizes calcium handling
PDE5 inhibitors	Vasodilation (smooth muscle cell relaxation)	Pulmonary hypertension	Inhibits NLRP3 inflammasome-mediated pyroptosis
		Erectile disfunction	
Phenofibrate (PPARα activator)	Increases lipolysis	Mixed dyslipidemia	Inhibits NLRP3 inflammasome-mediated pyroptosis
	Reduces apoprotein C-III		
Levosimendan (Calcium sensitizer)	Inotropic agent	Acute heart failure	Stabilizes calcium-troponin C complex
Istaroxime	Inotropic agent	Acute heart failure	Increases expression of SERCA-2 and sodium-calcium exchanger-1
SGLT-2 inhibitors	Increases urinary glucose excretion	T2DM	Lowers uric acid levels (oxidative stress)
	Lowers plasma glucose	Heart failure	Reduces epicardial adipose tissue and activin-A levels, which lowers inflammation and fibrosis
			Reduces left ventricular mass and fibrosis
			Increases oxygen release (hemoconcentration)
			Shifts toward free fatty acids oxidation
			Promotes ketogenesis

AMPK: 5′ adenosine monophosphate-activated protein kinase; GLP-1: glucagon-like peptide 1; NLRP3: NLR family pyrin domain containing 3; PDE5: phosphodiesterase type 5 inhibitor; PPARα: Peroxisome proliferator-activated receptor alpha; RC: respiratory chain; ROS: reactive oxygen species; SERCA-2: Sarco-Endoplasmic Reticulum Calcium ATPase-2; SGLT-2: sodium-glucose cotransporter-2; T2DM: type 2 diabetes mellitus.

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
