# Peer review of "Molecular Basis of Cardiomyopathies in Type 2 Diabetes"

_ijms, 2024, doi:10.3390/ijms25158280_

Round 1

Reviewer 1 Report

Comments and Suggestions for Authors

The Article submitted by Giardinelli et al. aims to underline the molecular basis of the pathogenetic mechanisms of cardiomyopathies in adolescents and young adults suffering from type 2 diabetes and to define future perspectives in relation to therapies.

In my opinion, the manuscript is well written but still requires further editing. The following points need to be addressed and clarified.

Major points:

-The Authors correctly mention some essential molecular protagonists in the pathophysiology of diabetes. What can they say about ERRalpha? Recent literature implicates it in the AMPK/PGC-1alpha/Sirt1 axis.

-The Authors are required to include a paragraph relating to phytohormones with particular attention to abscisic acid (ABA).

-What do the Authors think of UCP1/3 and ANT1 (ADP/ATP translocator)? They could insert them into the manuscript to improve it. Authors are therefore invited to read and possibly cite some articles in this regard (for example doi.org/ 10.3390/antiox12091692).

Minor points

-The Authors must correct various errors of form present in the manuscript and must include both the Authors' contributions and conflicts of interest if present.

-The Authors could cite some cell lines and some murine studies in greater detail.

Author Response

The Article submitted by Giardinelli et al. aims to underline the molecular basis of the pathogenetic mechanisms of cardiomyopathies in adolescents and young adults suffering from T2D and to define future perspectives in relation to therapies.

In my opinion, the manuscript is well written but still requires further editing. The following points need to be addressed and clarified.

Major points:

1.-The Authors correctly mention some essential molecular protagonists in the pathophysiology of diabetes. What can they say about ERRalpha? Recent literature implicates it in the AMPK/PGC-1alpha/Sirt1 axis.

ANSWER: We believe that pathophysiology of diabetes is beyond the scopes of this review, consequently we did not deep the ERRalpha and its role in the AMPK/PGC-1alpha/Sirt1 axis.

2.-The Authors are required to include a paragraph relating to phytohormones with particular attention to abscisic acid (ABA).

ANSWER: Phytohormones and particularly abscisic acid have been cited in literature as potential therapies for metabolic syndrome and type 2 diabetes; up to now, their role in diabetic cardiomyopathy has not been studied yet, so we believe that is beyond the aim of this review.

  1. What do the Authors think of UCP1/3 and ANT1 (ADP/ATP translocator)? They could insert them into the manuscript to improve it. Authors are therefore invited to read and possibly cite some articles in this regard (for example doi.org/ 10.3390/antiox12091692).

ANSWER: We thank the reviewer for the suggestion. To improve the manuscript, we have included the role of UCP1/3 and ANT1 ADP/ATP translocator in the DbCM (lines 153-177, references 56-60).

Minor points

-The Authors must correct various errors of form present in the manuscript and must include both the Authors' contributions and conflicts of interest if present.

ANSWER: done.

-The Authors could cite some cell lines and some murine studies in greater detail.

ANSWER: done, lines 70-72, 80-90.

Reviewer 2 Report

Comments and Suggestions for Authors

This study, "Molecular Basis of Cardiomyopathies in Type 2 Diabetes" was designed to highlight the increasing knowledge on molecular mechanisms of diabetic cardiomyopathy (DbCM) and the new therapies targeting specific pathways. The authors focused on the topic of (1) Inflammation, (2) Mitochondrial dysfunction, (3) Molecular mechanisms determining cardiac remodeling, (4)Myocardial calcium handling, (5)Epigenetic changes and (6)Future prospectives on the treatment in this review. This research covers a wide range and is very attractive. However, each topic's descriptions lack organization and are presented reference by reference, making it difficult to comprehend and concentrate on the main idea. Here are some suggestions that need to be improved.

(1)    Please summarize the findings in each topic/paragraph in a table with in vitro or in vivo findings, indicating the reference number. Figure 1 does not cover all topics; it must include the rest.

(2)    Typesetting problems, lines 39, 51,57, 154, 160, 164, 172, 184, 186, 194.

(3)    Lines 57-59, “The aim of this narrative review is to highlight the most recent evidence on the molecular basis of pathogenetic mechanisms of cardiomyopathies in adolescents and young adults affected with T2DM, and the future therapeutic prospectives.” The authors did not specific summary and compare the molecular basis of pathogenetic mechanisms of cardiomyopathies “in adolescents and young adults” affected with T2DM.

(4)    The authors should explain graphically the link or progression between apoptosis, autophagy, pyrolysis, cardiomyocyte loss in DbCM (lines 75-76), cardiac hypertrophy and fibrosis as a major feature of DbCM (line 144), as well as inflammation and metabolic stress contribute to senescence of both cardiomyocytes and immune cells (lines 101-103).

Author Response

Reviewer 2

This study, "Molecular Basis of Cardiomyopathies in Type 2 Diabetes" was designed to highlight the increasing knowledge on molecular mechanisms of diabetic cardiomyopathy (DbCM) and the new therapies targeting specific pathways. The authors focused on the topic of (1) Inflammation, (2) Mitochondrial dysfunction, (3) Molecular mechanisms determining cardiac remodeling, (4) Myocardial calcium handling, (5)Epigenetic changes and (6)Future prospectives on the treatment in this review. This research covers a wide range and is very attractive. However, each topic's descriptions lack organization and are presented reference by reference, making it difficult to comprehend and concentrate on the main idea. Here are some suggestions that need to be improved.

  • Please summarize the findings in each topic/paragraph in a table with in vitro or in vivo findings, indicating the reference number. Figure 1 does not cover all topics; it must include the rest.

ANSWER: We thank the reviewer for the suggestion. To better understand the topics discussed in the manuscript, we included Table 1 which contains the main molecular evidence, in vitro/ ex vivo/ in vivo findings, with the reference number. Figure 1 in the text has been improved.

  • Typesetting problems, lines 39, 51,57, 154, 160, 164, 172, 184, 186, 194.

ANSWER: done

  • Lines 57-59, “The aim of this narrative review is to highlight the most recent evidence on the molecular basis of pathogenetic mechanisms of cardiomyopathies in adolescents and young adults affected with T2DM, and the future therapeutic prospectives.” The authors did not specific summary and compare the molecular basis of pathogenetic mechanisms of cardiomyopathies “in adolescents and young adults” affected with T2DM.

ANSWER: there are not specific molecular pathways, so we changed the sentence in “The aim of this narrative review is to highlight the most recent evidence on the molecular basis of pathogenetic mechanisms of cardiomyopathies in T2DM patients, and the future therapeutic prospectives” 

  • The authors should explain graphically the link or progression between apoptosis, autophagy, pyrolysis, cardiomyocyte loss in DbCM (lines 75-76), cardiac hypertrophy and fibrosis as a major feature of DbCM (line 144), as well as inflammation and metabolic stress contribute to senescence of both cardiomyocytes and immune cells (lines 101-103).

ANSWER: We thank the reviewer for the suggestion. Figure 1 has been added to the manuscript that explains the relationship between apoptosis, autophagy, pyro-apoptosis, cardiomyocyte loss in DbCM, cardiac hypertrophy and fibrosis as a major feature of DbCM, as well as inflammation and metabolic stress contribute to senescence of both cardiomyocytes and immune cells.

Round 2

Reviewer 2 Report

Comments and Suggestions for Authors

The authors have responded to the reviewer's comments and have improved the manuscript. I have no additional comments.